# Autism Spectrum Disorders in Greece: Nationwide Prevalence in 10–11 Year-Old Children and Regional Disparities

**DOI:** 10.3390/jcm9072163

**Published:** 2020-07-08

**Authors:** Loretta Thomaidis, Nikoletta Mavroeidi, Clive Richardson, Antigoni Choleva, George Damianos, Konstantinos Bolias, Maria Tsolia

**Affiliations:** 1Developmental Assessment Unit, Second Department of Pediatrics “P. & A. Kyriakou” Children’s Hospital, National and Kapodistrian University of Athens School of Medicine, Athens 11527, Greece; nmavroidi@yahoo.gr (N.M.); antigonicholeva@gmail.com (A.C.); dimosthenisdamianos@yahoo.com (G.D.); kb@kbolias.gr (K.B.); mariantsolia@gmail.com (M.T.); 2Department of Economic and Regional Development, Panteion University of Social and Political Sciences, Athens 17671, Greece; crichard@panteion.gr

**Keywords:** Autism Spectrum Disorders, administrative data, prevalence, Greece, children

## Abstract

Autism spectrum disorders (ASD) constitute a public health concern with increasing prevalence worldwide. We aimed to estimate prevalence and age at diagnosis in Greece, where no large-scale prevalence study has ever been conducted. Aggregate data were collected on ASD diagnoses by gender and calendar year of diagnosis up to 2019, for children born in 2008 and 2009, from the Centers for Educational and Counseling Support, which evaluate children to receive special educational support in school. Coverage was 87.1% of centers and 88.1% of schoolchildren born in 2008–9. ASD prevalence overall was 1.15% (1.83% males, 0.44% females; ratio 4.14:1), ranging from 0.59% to 1.50% in Greece’s 13 regions. In five regions, prevalence differed significantly between centers. Overall, only 3.8% of diagnoses were made before the fourth year after birth and 42.7% before the sixth year, with considerable variation between regions. Approximate mean age at diagnosis was six years and one month, and about three months earlier for girls than for boys. Our results provide evidence-based information to guide service planning and development at national and regional levels. Particular attention should be paid to smoothing out inequalities regarding service accessibility and provision. Emphasis should be given to earlier identification and diagnosis of ASD.

## 1. Introduction

Autism Spectrum Disorders (ASD) are lifelong neurodevelopmental disorders, usually diagnosed during childhood, with an increasing prevalence worldwide over the last 20 years [1]. Individuals with ASD present with impaired verbal and non-verbal social communication and interaction, as well as restrictive or repetitive interests, behaviors or movements [2]. The severity of symptoms and the level of functioning vary, ranging from autonomous individuals who require only limited support, to persons with restricted autonomy, intellectual disability and other comorbidities, without verbal communication, for whom continuous and substantial support is necessary [3,4,5]. ASD-related challenges may differ across different phases of life, leading to multiple and changing needs over the lifespan of individuals with ASD and their families. The diagnosis of ASD is based on clinical criteria, as formulated in the International Classification of Diseases (ICD-10) of the World Health Organization or the Classification of the Psychiatric Disorders (DSM-5) of the American Psychiatric Association [2,6]. Genetic, epigenetic and environmental risk factors have all been implicated in the occurrence of the disorder [7,8,9], but a recent analysis of large Swedish twin cohorts reported that genetic factors consistently played a larger role than environmental ones [10]. In most cases, the etiology remains unknown.

Up until the early 1990s, autism was included among the rare diseases, namely those presenting with a prevalence of less than 5/10,000 [11]. Nowadays it is well known that autism is definitely not rare, with an average prevalence of 0.90% in North America and 0.61% in Europe [12]. The median prevalence worldwide was estimated at 0.7% between 2000 and 2016 [13,14]. ASD prevalence estimates from the majority of recent studies cluster in the range of 0.8% to 1.5%, while prevalence has constantly been on the rise during the last 15–20 years [15]. During the same time period, the prevalence across the 11 sites of the Autism and Developmental Disabilities Monitoring (ADDM) Network in the USA has increased by 2.8 times, reaching 1.85% in 2016. Increasing sensitivity of the diagnostic criteria, more effective and earlier diagnosis, better surveillance of epidemiological data and increased awareness on the part of health professionals, teachers and parents, are all factors potentially involved in the observed steady increase in ASD prevalence [16]. 

There is a marked gender difference in the prevalence of diagnosis of ASD, with males more frequently affected than females. A meta-analysis of 54 studies from 1992–2011 estimated the overall male-to-female odds ratio to be 4.20 (95% confidence interval 3.84–4.60), although a little lower at 3.32 (2.88–3.84) in the subgroup of 17 studies regarded as high quality [17]. Females tend to present with less obvious symptoms and signs compared to males [2,12,13,18]. The gender difference appears to be smaller among children with moderate to severe intellectual disability [3,11].

Increasing demand for early detection has been observed in parallel with the rise in ASD prevalence [19]. The focus on early diagnosis is driven by emerging evidence that early intervention (from birth to 36 months) can substantially improve a child’s development, conferring significant benefits in cognitive skills, adaptive behavior, communication and overall functioning [20,21]. Furthermore, evidence nowadays suggests that ASD can be diagnosed reliably before two years of age, the diagnosis made after 14 months being stable up to 3–4 years of age in 84% of the children in a large community sample [22,23]. However, despite the global trend towards earlier diagnosis, recent studies demonstrate that considerable disparities still exist with regard to the age at diagnosis. Between 1990 and 2012, the median age at diagnosis reported in the literature ranged from 34 to 88 months. Recent estimates include 4.25 years in the USA [18] but 8.8 years in Catalonia [12]. Factors associated with earlier diagnosis include more severe symptoms, lower functioning, higher socioeconomic status and increased parental concern. In addition, family interactions with the health and education systems prior to diagnosis, and racial and ethnic factors can also influence the age at diagnosis [24,25]. 

In the last decade, both the number of ASD prevalence studies and the number of reporting countries have increased. A variety of methodologies and populations have been used, depending on the existing infrastructure and organization of each country, the availability and accessibility of data, and resources. Methodological differences impose the need for caution when comparing prevalence estimates between countries and studies. In addition to disparities in methodology, the differences between studies and countries might also be attributable to cultural and environmental factors [15,26]. However, regional prevalence has been reported to differ within the same country, even in whole population designs [12,27,28,29,30,31,32,33]. The use of different diagnostic procedures and tools, awareness of health professionals, accessibility to available diagnostic services and specialists, and organization of services have been implicated in regional and geographical disparities [18,24,34].

The vast majority of ASD prevalence studies in the literature can be classified into one of the following four categories, with regard to the methodology used and the source of the data: 

Use of administrative data from national registries, birth cohorts, health records and electronic health files. In general, birth cohorts are popular in Scandinavian countries, such as Sweden [35] and Denmark [36]. Health sector administrative data have been employed in many studies, including Denmark, Finland, France and Iceland within the Autism Spectrum Disorders in European Union (ASDEU) project [36] and in Catalonia, Spain [12]. 

Screening and diagnosis in a large sample of the school population. Examples in this category include studies conducted in the UK [37], Ireland [38], Spain [39], South Korea [40], Taiwan [41], China [27] and Mexico [42], among others. 

Recording of reported ASD diagnosis from the parents or teachers. This methodology was employed in the National Survey of Health in the USA [43,44], and in studies in Sweden [35], Spain [45] and Scotland (using a question included in the 2011 Census [46]).

Estimation of prevalence in a sample of the population, using primary data from health and education records with subsequent confirmation of the diagnosis, in the USA [18,47]. 

Although the references above cite many European studies, it may be noted that few of these refer to Mediterranean Europe and Southern Europe in general, with the exception of some studies in Spain [12,39,45,48], Italy [49,50,51] and Southern France [36]. In particular, no large-scale epidemiological study of ASD prevalence has been conducted in Greece previously. The increasing prevalence of ASD, in association with its lifelong duration and the nature and severity of the presenting challenges and comorbidities, determines the extent and the content of the investment required in planning and developing the necessary resources and services for individuals with ASD, in the areas of public health, education and social care. The purpose of the present study is to provide epidemiological data on ASD in Greece, in order to support the evidence-based planning and development of the necessary resources and services, nationally and regionally. Specifically, our objectives were to estimate ASD prevalence and age at diagnosis in Greece, and to identify regional disparities in a whole country population, in children 10 and 11 years old. 

## 2. Population and Methods

### 2.1. Population 

Prevalence was estimated among children in the final grades of primary school, by year of birth and calendar year at diagnosis, as previously suggested in the literature [52,53]. Specifically, we estimated the prevalence of autism nationwide in Greece among children who were born in 2008 and 2009, thus reaching the ages of 11 and 10 years in 2019. 

### 2.2. Source of the Data

We obtained the numbers of cases of autism in children born in 2008 and 2009 from the Centers for Educational and Counseling Support of the Ministry of Education. These centers evaluate students with special educational needs (SEN) and provide support and counseling to students, their families and the schools, with the aim of promoting inclusive education and the optimal integration of every child into the indicated educational setting. Every school in Greece (public or private, regular or special) is allocated, based on its location, to a center. Thus, every child is subject to only one center, according to the school attended for each academic year, the child’s file being transferred in case of change of residence.

Evaluation by the center which covers the area of the child’s school, is the sole route to the child’s classification into one or more of the categories of SEN defined by the Ministry of Education. One such category refers to SEN related to the diagnosis of “Autism”: “pervasive developmental disorder” according to ICD10 or “Autism spectrum disorders”, according to DSM-5. A child can only be categorized as “SEN Autism” based upon a clinical diagnosis issued by a public sector child psychiatrist, child neurologist or developmental pediatrician. 

Although in 2019 a total of 71 centers had been established by law in the 13 administrative regions of Greece and the 54 Regional Entities into which these are divided, several of them were projected future centers to be formed by splitting the areas covered by existing centers in large cities. Thus, at the time of the study, all of Greece was covered by 62 centers. All centers are located in the capitals of the Regional Entities.

The selection of this particular source of administrative data was based on the availability and accessibility of the number of cases, in association with the availability of well-defined source populations, categorized by year of birth and grade, to be used as denominators for calculating prevalence. The populations served by each center, by gender and by year of birth (2008 and 2009), were provided by the Ministry of Education from its administrative data on schools. Further data on the populations of these areas were obtained from publications of the Hellenic Statistical Authority.

### 2.3. Data Collection

All 62 centers were invited to participate in the study. Data were collected by the designated staff of each center, under the responsibility of its Director, using the data collection tool provided by the research team. Written guidelines and guidance were provided, with FAQs, as well as videotaped material, in order to support the retrieval procedure and data collection from the paper or digital files or databases. File keeping and the organization of the archives differ substantially among the centers and vary from the exclusive use of paper files in some cases to different types of computerized databases in a minority. The majority had a combination of paper and computer-based files and listings. Tailored guidance was provided according to the type of file and archive organization. Every center was contacted at least twice by the research team during the data collection period in order to support the collection procedure and minimize errors and biases in collection. The study was carried out in June and July 2019, with data collection completed in February 2020 only for two centers.

### 2.4. Data Collection Tool

A study-specific questionnaire was created for the needs of the study. Aggregate anonymous data were reported on the number of autism cases, by year of birth (2008 or 2009), gender and year of first diagnosis. Data collection by calendar year was expected to be important in increasing the response rate by reducing the response burden upon centers.

### 2.5. Case Definition

A student born in 2008 or 2009, evaluated in the center between 1 January 2012 and 20 July 2019 (the closure date of the Centers for the academic year 2018–19), with special educational needs related to a diagnosis of autism-pervasive developmental disorders or autism spectrum disorders recorded at any point of the child’s life, irrespective of other concomitant diagnoses or other categories of SEN. 

### 2.6. Age at Diagnosis

Diagnoses were recorded by calendar year (2011–2019), not by the child’s actual age. The first year of recorded diagnosis was the third year after birth, that is, 2011 for children born in 2008, and 2012 for those born in 2009. These children all reached the age of 3 in that year, but with a potential range of age at diagnosis from 2 years and 1 day to 3 years 11 months and 30 days depending on the exact dates of birth and diagnosis. Similarly, children who were diagnosed in the fourth year after birth (2012 if born in 2008 and 2013 if born in 2009), reached the age of 4 in that year and had age at diagnosis of 4 ± 1 years, and so on.

### 2.7. Statistical Analysis

Methods appropriate for count data were employed for the statistical analysis of the data. Point prevalence was calculated using the number of ASD cases in the numerator divided by the source population in the denominator, at the national, regional or center level as appropriate and stratified by gender and year of birth where required. Percentages and frequency distributions were compared using chi-squared tests. Poisson regression was used for the multivariate analysis of rates of ASD diagnosis in relation to gender, year of birth and location (region or center) simultaneously. This was carried out on national data in order to examine differences between regions, followed by separate analyses within each region in order to examine differences between centers. Because diagnoses in year 11 could be recorded only for children born in 2008, we excluded year 11 from the analyses of age at diagnosis.

### 2.8. Ethics

Data were reported anonymously by the centers as aggregated numbers of autism cases by gender, by year of birth and by year of first diagnosis, under the responsibility of the Director of the center. The data were accessible only to members of the research team and were used exclusively for scientific purposes within the aim and objectives of the study. Ethical approval for the study was granted by the Bioethics and Ethics Committee of the National and Kapodistrian University of Athens (Reference Number 116/15-04-2019). Permission to conduct the study was granted by the Department of Special Education of the Hellenic Ministry of Education, Research and Religious Affairs as the authority responsible for the Centers for Educational and Counseling Support (Reference Number 65351/Δ3/23-04-2019).

## 3. Results

Fifty-four of the 62 Centers for Educational and Counseling Support participated in the study, corresponding to a study population of 182,879 schoolchildren born in 2008 and 2009 out of the national total of 207,660. Thus, the coverage was 87.1% in terms of centers and 88.1% in terms of population. Of these 182,879 children, 92,321 and 90,558 were born in 2008 and 2009 respectively, and 51.3 % were male (51.2% in 2008 and 51.4% in 2009). The range of response rates in the 13 administrative regions of the country was 63–100% of centers, with a coverage of 78.6–100% of the population. Coverage was 100% in 7/13 regions.

In total, 2108 diagnoses of autism in children born in 2008 and 2009 were recorded, with an overall prevalence of 1.15% at the national level (1.18% and 1.13% in 2008 and 2009, respectively; 1.83% in males and 0.44% in females, male-to-female ratio 4.14:1 in terms of prevalence, 4.36:1 in terms of numbers of diagnoses). The gender-specific prevalence and the male-to-female ratio did not differ between 2008 (1.87% for boys vs. 0.44% for girls, 4.33:1) and 2009 (1.78% vs. 0.44%, 4.05:1). Treating the achieved sample of children as a simple random sample of the population, and applying a finite population correction, 95% confidence intervals for the population percentage prevalences were 1.80–1.86 for boys, 0.43–0.46 for girls and 1.14–1.17 overall. The prevalence in each administrative region is presented in Table 1 for the total population and broken down by gender in Table 2. Overall prevalence by region is shown in Figure 1. The regional prevalence ranged from 1.50% in the North Aegean to 0.59% in Western Greece and differed significantly between regions (*p* < 0.001). Prevalence was also high (1.46%) in the Attica region, which contains the major population center of Greece, the conurbation of Athens and Piraeus. Eight of the other 12 regions had a significantly lower prevalence than Attica (*p* ≤ 0.001) in the Poisson regression, the exceptions were the islands of the North Aegean, the Ionian Islands, Crete and Eastern Macedonia and Thrace. Significant differences were not found between the years of birth 2008 and 2009 in this or any other analysis (*p* > 0.05) and this factor will not be mentioned again.

Whereas the gender-specific prevalence differed between males and females within every region (*p* < 0.001), the male-to-female ratio did not show a statistically significant difference (*p* = 0.41) even though it ranged from 2.8 in the Peloponnese to 10.1 in the North Aegean. Seven regions presented with a male-to-female ratio greater than 4:1 and only two lower than 3:1.

In individual centers, prevalence ranged from 0.31% to 2.59%, with a median of 1.01% (interquartile range 0.71% to 1.41%). Prevalence at this geographical level is shown in Figure 2. In an attempt to examine whether local prevalence was associated with difficulty of access to the centers, we calculated the correlation between the overall prevalence and the percentage of rural residents in the population in the center’s area of responsibility, and also between the prevalence and the percentage of population living in mountainous areas. Both calculations excluded the major cities (Athens, Piraeus and Thessaloniki). We found that neither correlation was statistically significant (Pearson correlation *r* = 0.22, *p* = 0.15 for the rural population; *r* = 0.14, *p* = 0.39 for the population in mountainous areas).

In the Poisson regression analyses by gender and center separately within each region, the gender effect was significant in every case (*p* < 0.001) and did not differ between centers (*p* > 0.05). There was a statistically significant difference between centers (*p* ≤ 0.003) in five regions: the Ionian Islands, Crete, Central Macedonia, Western Greece, and Eastern Macedonia and Thrace. 

Frequencies and cumulative frequencies of ASD diagnoses are shown by year of diagnosis and gender in Table 3. Overall, only 3.8% of cases were diagnosed before year 4 and fewer than half (42.7%) before year 6, which is the year of starting primary school. Most diagnoses (46.3%) were made in years 5 and 6, and the great majority (88.8%) had been made before and up to the end of year 8. The distribution of the age at diagnosis differed between boys and girls (*p* = 0.007) and was particularly large in year 5, when 30.3% of girls’ diagnoses were made compared to 24.4% of boys’. Because diagnoses were recorded by calendar year, it is not possible to calculate exactly the mean age at diagnosis. However, making the assumption that dates of birth and dates of diagnosis are distributed evenly throughout the year, rough estimates of 6 years and 2 months for boys, 5 years and 11 months for girls and 6 years and 1 month overall are obtained.

Table 4 shows ASD diagnosis prevalence at each year up to the tenth after the child’s birth. Overall prevalence was 0.49% in year 5, which was the year with the lowest male-to-female ratio. The gap between males and females widened steadily thereafter.

Findings on the year of diagnosis in each region are presented in Table 5 (mean age at diagnosis and modal year) and 6 (cumulative frequencies by year). Differences in distribution between regions were statistically significant (*p* < 0.001). The correlation between regional prevalence and mean age at diagnosis was not significant (Pearson correlation—0.13, *p* = 0.68). The possibility of comparing age at diagnosis between centers within regions was limited by the small number of diagnoses in the majority of them; 38/54 (70.4%) had fewer than 25 cases. In the region of Attica, where the six participating centers had between 65 and 214 cases, the age distribution did not differ between centers (*p* = 0.54).

## 4. Discussion

In our study we assessed the ASD diagnosis prevalence in 2019 and the age at diagnosis in an entire country’s population of 10- and 11-year-old children (born in 2008 and 2009), using administrative data from the public Centers for Educational and Counseling Support in Greece. The overall prevalence of autism diagnosis in this population was estimated to be 1.15% (1.83% in males and 0.44% in females; male-to-female ratio 4.14:1, with no difference between the two years of birth in respect of the overall and gender-specific prevalence or the male-to-female ratio). No previous estimates exist from large-scale epidemiological studies in Greece. The only existing estimate of ASD prevalence of which we are aware is 0.32% found in a local study in primary schools in part of the island of Crete in 2012, in which the diagnosis of autism was taken from school records [54]. 

The average prevalence of ASD in Europe across 12 countries taking part in the ASDEU study was 1.22% [55], compared with a median of 0.61% estimated in a meta-analysis a few years earlier [13]. The results of our study may be compared in particular with recent European studies that, like ours, employed administrative data in a whole country population design: Norway 0.7% [30], Finland 0.76% [36], Norway 0.9% [56], Iceland 1.2% [57], Denmark 1.26% [36] and Sweden 1.7% at ages 6–12 years and 2.5% at 13–17 years [58]. More recent results for Iceland come from the ASDEU study in which the national ASD prevalence was 3.13%, apparently the highest reported from any country. The authors attributed the substantial increase over the previous estimate to significant changes in the country’s diagnostic services and increases in the available resources after 2010 [36]. By also using administrative data, the National Autism Surveillance system in Canada estimated prevalence of 1.52% in 2015 for children from 5 to 17 years old [34]. 

Administrative data from health services have also been used in European studies implemented in specific regions, giving prevalence estimates of 1.23% in Catalonia, Spain [12]; 0.95% in L’Aquila in Central Italy and 1.04% in the 10–11 age group [59]; 0.94% in Cambridgeshire, UK [37]; 0.86% in Pisa, Italy, increasing to 1% when the results from the screening part of the study were added to already established diagnoses [51]; 0.48% in South Italy [49]; 0.48% in south-eastern parts of France and 0.73% in the south-west [36]; and 0.35% in the West Pomeranian and the Pomeranian regions of Poland [60]. Although the above studies employ basically the same study design (based on the use of administrative data), they differ substantially with regard to the age range and the number of birth cohorts included in the population, and the time period used for cumulative prevalence calculation or point prevalence estimation. As we also observed in our study, geographical and regional disparities are frequent in whole country population designs, thus limiting the conclusions that can be drawn when comparing results between nationwide and regional studies.

Results based on already established diagnoses from administrative data, without additional confirmation or ascertainment of diagnoses, are likely to be influenced by several factors: the diagnostic procedures and tools in predominant use; the availability and accessibility of diagnostic services in the community; awareness and clinical experience of health professionals; the type of ASD diagnosed; and by the reporting procedures. These factors account for the introduced biases and observed differences among studies that should be interpreted with caution [1,12,27]. Our study bears particular similarity with that implemented in Pisa, Italy [51], as both studies retrieved cases from the local services of the Ministry of Education. However, we neither confirmed the diagnosis nor collaborated with the health services that had originally issued the diagnoses, as the Italian research team did.

Another widely used study design involves a two-step procedure of screening and diagnosis. One of the higher prevalence estimates in Europe was recorded using this design in a general population sample in Spain: 1.55% in preschool children and 1% in school age children [61]. Similar designs had previously been used in the UK (prevalence 1.57%) [37] and in Pisa, Italy (0.3%) [51]. Outside Europe, a 2.64% prevalence was reported using a similar design in South Korea [62], although those results were eventually considered as biased, leading to an overestimation of the prevalence in a highly assumption-dependent design accounting for the low participation rate [40]. In China, an estimated prevalence of around 1% in three different regions and in Beijing was found using screening, clinical assessment and diagnosis in the population of mainstream and special schools [63,64]. Studies using a screening and diagnosis design are known to identify children not previously diagnosed (35–50% of cases), who may tend to have milder or high-functioning cases, whereas more severe cases are likely to have an established diagnosis already [37,51,61]. However, surveys of this type are also subject to various biases related to the validity of the case definition and diagnosis, and to the diagnostic procedure and screening tools used, while a low participation rate may constitute one of their major challenges [1,27,40].

The 11-site ADDM network in the USA has been systematically estimating the prevalence in 8-year-old old children every two years since 2000, screening and ascertaining the ASD diagnosis from the health and educational records of the study sample. The most recent prevalence published by this network was 1.85% in 2016 [18], compared to 1.68% for 2014 [33]. While the screening and diagnosis procedures are not independent and direct case assessment is not required for case confirmation, these Centers for Disease Control and Prevention (CDC) studies provide useful information on the ASD prevalence time trend in this population [1].

However, based on parental reports obtained from the interviews held within the National Health Interview Survey, higher prevalence estimates have been obtained in the USA: 2.34% [43] and 2.47% [65]. Prevalence estimated from information provided by parents has also been reported for Scotland from the 2011 National Census data analysis (1.9% in the population 0–17 years) [46]. Using a telephone interview screening with parents in Sweden, the child autism phenotype prevalence was estimated to be 0.95% [35]. In Australia, ASD diagnosis prevalence based on parental reports was 3.9% and 2.4% for two different birth cohorts aged 10–11 years old [66]. Prevalence studies based on parental or teachers’ reports are subject to significant interviewer or questionnaire-related biases [1,43]. The association between prevalence estimates and source of information is investigated in a review of recent prevalence studies [67]. 

Males were more frequently diagnosed than females in our study, as is universally noted in the literature. Our overall male-to-female ratio of 4.14:1 is close to the ratio of 4:1 cited in the DSM-5 Manual [2] and the value of 4.2 found in a meta-analysis of 54 studies [17], as well as the results of recent studies: USA 2016 4.3:1 [18], Catalonia 4.5:1 [12], Spain 4:1 [61] and Sweden 3.3:1 [58]. In research based on administrative data, the ratio ranged from 3.3 to 5.4 in the ASDEU studies [36] and was almost 5:1 in Pisa [51]. Several factors have been implicated in this well-established male excess. Although the vulnerability or protective effect of gender chromosomes is not clear, biological evidence supports the role of sex hormones, specifically fetal testosterone, on the genetic predisposition for autism, and on behavioral and communication characteristics and autistic traits [68,69]. Recent studies suggest that girls present with different structural brain alterations related to autistic traits, compared to boys [70]. It seems that the “female ASD phenotype” is often different from the prevailing male one, especially among high functioning individuals. Females are considered to be less likely to manifest restricted interests or overt communication challenges, and have been described as camouflaging their difficulties, leading to escaping or delaying the diagnosis [71]. Even when autistic traits exist, they often fail to meet the established diagnostic criteria, which are considered to probably lack equal sensitivity for the male and female phenotypes, leading to a diagnostic bias [72]. This bias might be further enhanced by gender stereotypes that are potentially held by the health professionals who make the diagnosis, in the sense that they are probably more predisposed to consider autism for a boy than for a girl. In addition, parents and teachers express substantially fewer concerns about girls than boys [73]. Finally, while the male predominance is not in dispute, the quality of studies and their design may influence the size of the gender ratio in autism. The authors of the 2017 meta-analysis pointed out that earlier evidence was drawn from isolated studies with widely varying designs, whereas the studies that they regarded as being of better quality, with direct ascertainment of the diagnosis (screening and diagnosis in the general population), yielded lower male-to-female ratios of around 3.3 compared to 4.2 in all studies [17]. In common with many studies, we were unable to examine whether the male-to-female ratio varied according to the level of intellectual functioning; there is evidence in the literature that it may be relatively low, around 2:1, among children with ASD and moderate-to-severe intellectual disability [3,11].

As evidence emerges about the positive impact that early interventions starting before the third year of age may confer on the behavior and communication skills of toddlers, interest is increasing regarding the age at which the diagnosis is made [19]. Although the mean age at diagnosis for eight-year-old children has been reported to remain stable at around 55 months in the USA for the last 10 years [74], several studies suggest that it can vary from 36 to 120 months, with many children remaining undiagnosed until school age or even until 9–11 years for milder cases, such as Asperger syndrome and Pervasive Developmental Disorder NOS [24,75,76,77]. On the other hand, children with prominent symptoms such as language regression and more severe presentation, with autistic disorder and intellectual disability, are diagnosed earlier [25,78,79]. In Sweden, where the mean age at diagnosis was eight years, older age at diagnosis was correlated with a lack of intellectual disability [80]. Similarly, as reported by the USA ADDM network, the median age at diagnosis in eight-year-old children in 2016 was 51 months for cases with IQ ≤ 70 but 57 months for those with IQ > 70 [18]. 

In the absence of language delays, cognitive and adaptive challenges may go unrecognized until the environmentally expected social communication skills overwhelm the abilities of the child, thus delaying the diagnosis until adolescence or early adulthood [24]. Finally, applying time-to-event analysis to the data obtained from the USA National Survey of Children’s Health 2011–2012 and from the National Survey of Children with Special Health Care Needs 2009–2011, only a minority of children had been diagnosed before the age of three and between one third and half of them had received a diagnosis after six years of age, underlining the impact of the study design and introduced sampling biases when estimating the age at ASD diagnosis [81].

The mean age at diagnosis was 8.8 years (median eight years) in a study population aged 0–17 using administrative data in Catalonia Spain in 2017 [12], compared to the estimated mean age of six years and one month in our study. In addition, almost half of the ASD cases in Catalonia (48%) had been diagnosed in the 6–10 years age group and 30% in the 2–5 years age group, compared to 57% and 42.7%, respectively, in our study. In a study population aged 3–17 in Central Italy, 50% of the children aged 9–11 years had been diagnosed at age 3–5 [59], compared to our 42.7%. From Spain, a study using a two-step screening and diagnosis procedure, reported a mean age at diagnosis of three years and 9 months in preschool children and seven years in children aged 10–12 years [61], while 80% of our cases had been diagnosed before the age of eight. Finally, seven years was the mean age at diagnosis reported in a study from Japan based on administrative data [82], with more diagnoses occurring at age three compared to our study. Different environmental expectations between Greece and Japan in the preschool setting with regard to the children’s behavioral and communication abilities, or different screening and early intervention service provision, might account for the pronounced disparity between the two countries.

The proportion of diagnoses made up to and including the third year was particularly low in our population (3.8%); at this stage the prevalence was only 0.04%, increasing to 0.20% the following year and 0.49% in the fifth year. We agree with previous authors that low prevalence at young ages might be related to a number of factors: low parental concern related to poor recognition of behavioral or communication challenges [19,83]; limited access to medical evaluation or awareness about early intervention services; developmental deviations raising concern for unidentified ASD or delayed referral if identified; low awareness of physicians regarding ASD presentation at early ages, or potential reluctance to diagnose a young child as autistic in order to avoid perceived stigmatization [12,82,84]. In Greece, many infants and toddlers are followed regularly by a pediatrician who monitors the development of the child and the acquisition of developmental milestones in the first years of life. However, potential disparities with regard to access to care might be related to membership of ethnic minorities, family and community socioeconomic status and parental education level, as previously described in other countries [18,80,85,86] as well as in Greece [87]. In addition, Greece is a country where the severe economic crisis of the last decade has imposed austerity that is observed to have had an impact on access to healthcare [88]. An attempt has been made in Greece to introduce screening tests for autism, through the distribution of M-CHAT [89] with the Child Health Booklet that is used for monitoring a child’s development and vaccination schedule, and by universities’ training of pediatricians and special education teachers in the “PAIS” screening test for communication disorders in toddlers (18–48 months) [90,91,92]. However, evidence is lacking on the extent to which pediatricians are actually using these screening tools at 18 months, or any other systematic screening procedure later in life. Nevertheless, beyond any failures in detection of ASD cases, additional factors which account for delayed ASD diagnosis have been identified from studies focusing on children ultimately diagnosed at five or six years of age, after having tested negative for ASD at age three or four. These include diagnostic overshadowing (previous multiple diagnoses), symptoms increasing over time or with delayed onset within the continuum of onset timing in ASD and sub-threshold ASD signs at age three evolving into impairment later when environmental demands exceed the child’s abilities [74,93]. 

The distribution of age at diagnosis was similar between the two years of birth in our cross-sectional study, whereas different distributions have been reported in previous studies, mainly with a longitudinal design and across a wider range of years of birth. An increasing proportion of diagnosis acquired at 3–5 years was observed between children born in 2001–03 and 2010–12 in Central Italy [59]. In the same direction, the cumulative incidence by 48 months of age was higher for children aged four years compared to those aged eight years in the USA ADDM network in 2016, pointing to an earlier age of ASD identification for the former group [18].

In contrast to results from Japan, which did not show a difference in age at diagnosis between genders [82], the distribution of age at diagnosis differed between boys and girls in our study, with significantly more girls than boys receiving early diagnosis, especially in year 5. At this point, the male-to-female ratio was at its lowest—3.46:1—remaining below 4:1 until year 9. Taking into account that delayed diagnosis tends to occur in milder ASD cases, with a higher IQ [18,80], particularly in girls without mental retardation or pronounced behavioral problems [2,94], we consider that the observed difference potentially refers to girls with more severe symptoms than boys, although data about functioning and intellectual disability were not available in our study.

We observed significant disparities with respect to overall and gender-specific prevalence across the 13 administrative regions and the 54 participating centers (Figure 1 and Figure 2, Table 1 and Table 2) as well as to the age at which diagnosis was made. Similar pronounced geographical disparities have been described in previous studies. In the USA ADDM network, prevalence ranged from 1.31 to 3.14% across the 11 sites in 2016 [18]. The National Autism Surveillance System in Canada reported a variation of 0.8–1.75% across provinces and territories [34], while another study in the same country identified marked differences in teacher-reported prevalence at the level of neighborhoods in an entire country population study [32]. As already noted, parts of South-Eastern and Southwestern France presented notably different prevalence estimates in the ASDEU study, attributed to marked differences with respect to service provision [55]. Similarly, prevalence differed significantly between healthcare areas in Catalonia, Spain, ranging from 0.55 to 1.84% [12]. In the same study, healthcare areas also differed with regard to the male-to-female ratio, although within a narrower range (4.1–5.5) than we observed among regions in our study (2.8–10.1). While prevalence was not correlated with age at diagnosis in our study, it is noticeable that the three regions with the lowest numbers of ASD diagnoses also had the lowest mean age at diagnosis, which might suggest a tendency towards detecting mainly the, presumably more severe, cases with early presentation. In agreement with this, Epirus and Western Macedonia had two of the three lowest prevalences of all the regions (Table 2). However, the third of these regions with low mean age at diagnosis, the North Aegean islands, had the highest prevalence of any region. Furthermore, the region of Western Greece was the lowest in prevalence but the highest in mean age at ASD diagnosis. A striking feature of the age distributions is the very large difference between Greece’s two major urban areas, Attica (Athens-Piraeus) and the region of Central Macedonia (Thessaloniki). Whereas about three-quarters of diagnoses were made up to and including year 6 in Attica, a further two years were needed to reach this level in Central Macedonia (Table 6).

In agreement with other authors, we consider that the geographical disparities observed in our study between regions and between centers within regions are probably related to differences regarding the diagnostic procedures and tools used, as well as the availability and accessibility of services and specialized professionals with clinical experience in ASD. The diagnoses provided to the centers had been made by specialist physicians in the public sector, namely, child psychiatrists, developmental pediatricians or child neurologists, the ICD10 criteria being used for coding the cases within the social insurance system. The gold standard tools ADOS and ADI-R are known to be employed and training programs are organized for professionals; however, while the diagnosis depends on the experienced clinical evaluation and expertise of health professionals [1], it is not known to what extent these tools are applied, the procedure is not fully standardized and the availability of services and experienced professionals seems to vary between the areas covered by the centers. The significantly higher ASD prevalence that we found in Attica compared to most other regions might be due partly to the concentration in the area of the nation’s capital city of specialists to make the diagnosis and other health professionals to make the referral. The fact that the data were reported by the local education services may have introduced an additional reporting bias: whereas the centers all use the same certification categories of special educational needs as defined by the Ministry of Education, they differ substantially in respect to organization and mode of record-keeping, which potentially influences the accessibility and reporting of the data.

Our study is the first epidemiological study in Greece to estimate ASD prevalence and age at diagnosis, nationwide or even on a large regional scale. It provides evidence-based information that can be used to estimate the burden of the disorder and guide service development and planning at national and regional levels. As mentioned in previous studies [12,36], the use of previously established ASD diagnoses from administrative data constitutes a reliable and cost-effective approach for estimating the ASD prevalence. The strengths of our study include a large study population of 182,879 children aged 10 and 11 years old, with a high population coverage of 88.1% and a high response rate with respect to the source of the data, as 87% of the Centers for Educational and Counseling Support participated in the study at the national level. The services of these centers are open to all children for the evaluation of special educational needs, from both public and private schools, without any institutional barriers to access. Furthermore, a child can only obtain special educational support after the local center has evaluated his or her needs. The age of the children is also considered to be a strength of our study: ASD prevalence is known to increase with age up to adolescence, thus the prevalence in 10–11-year-old children allows for estimates close to the real burden of the disorder in the community. In addition, we have been able to identify the distribution of ASD cases by age at diagnosis in our study population, which provides valuable information about when diagnosis is made and the differences between genders and among regions, thus enabling the targeted planning of service development with respect to referral, diagnosis, early intervention, rehabilitation, social care and social integration. 

We acknowledge several limitations in our study. The use of administrative data means that the cases are restricted to already-established diagnoses, which implies a possible underestimation of true prevalence. In addition, the quality of our data depends on preceding diagnostic procedures, on the availability and accessibility of diagnostic services, on the awareness of parents, teachers and health professionals, on the expertise of physicians in charge for screening and diagnosis and on the organization and file-keeping particularities of the counseling centers. Furthermore, we have not proceeded to additional confirmation of the established diagnoses, although the necessary degree of independence would not have been ensured between screening and diagnosis procedures, in order to prevent the introduction of additional information bias [1]. In addition, we have controlled neither for family nor the community socioeconomic status, which could potentially account for the observed geographical disparities and variance, as noted in previous studies [86]. Besides the lack of family data, our study design also precluded collection of data on children’s IQ.

A limitation imposed by our study design is its inability to identify diagnosed ASD cases that did not reach the centers, most probably milder cases or high functioning individuals, without prominent special educational needs, as noted in previous studies [37,51,61,66,84]. Furthermore, we were unable to calculate the mean age at diagnosis accurately, as cases were collected based on calendar years; however, we obtained accurate information on the full distribution of ages at which children are diagnosed, which is necessary for service planning and development of human resources. Despite its limitations, retrieving the data from the centers was considered to be the only feasible procedure in the absence of an established ASD register or other accessible health records associated with defined reference populations that could be used in the denominators. 

## 5. Conclusions

Our study is the first nationwide epidemiological study to estimate ASD prevalence in Greece. The prevalence in children aged 10 and 11 years old in 2019 was 1.15%, with boys affected more frequently than girls (ratio 4.14:1). The mean age at diagnosis was six years and one month and cases were most frequently diagnosed in the fifth calendar year following the year of birth. We observed marked disparities in prevalence and age at diagnosis at the local and regional level, even among major urban areas. Our results provide evidence-based information that can be used to estimate the global burden of the disorder in the community and guide service planning and development at the national and regional level, so as to meet the needs of people with ASD during their lifespan. Emphasis should be given to earlier identification and diagnosis of ASD cases, in association with the availability of accessible early intervention and rehabilitation services. Systematic screening of toddlers and older children, increasing awareness of parents and education and health professionals (especially pediatricians), and ensuring referral to centers specializing in ASD staffed with experienced clinicians, can all contribute to increasing the identification of ASD cases at an earlier age. Special attention should be paid to smoothing out inequalities at the local and regional levels with regard to service accessibility and provision. Standardizing the case-recording procedures across the education and health services and developing an ASD register at the national and regional level would promote the systematic monitoring of ASD epidemiology in Greece and evaluation of available services and allocated resources.

## Figures and Tables

**Figure 1 jcm-09-02163-f001:**
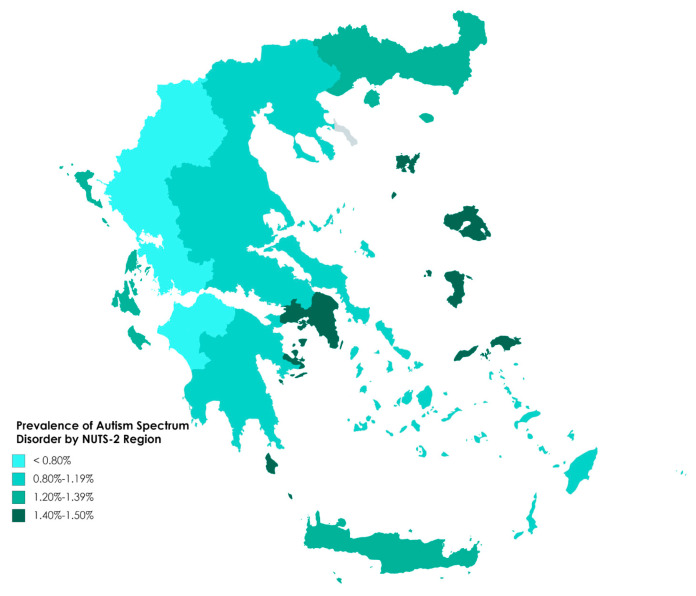
Prevalence of Autism Spectrum Disorder in the administrative regions of Greece (corresponding to the European Union’s NUTS-2 level of territorial classification).

**Figure 2 jcm-09-02163-f002:**
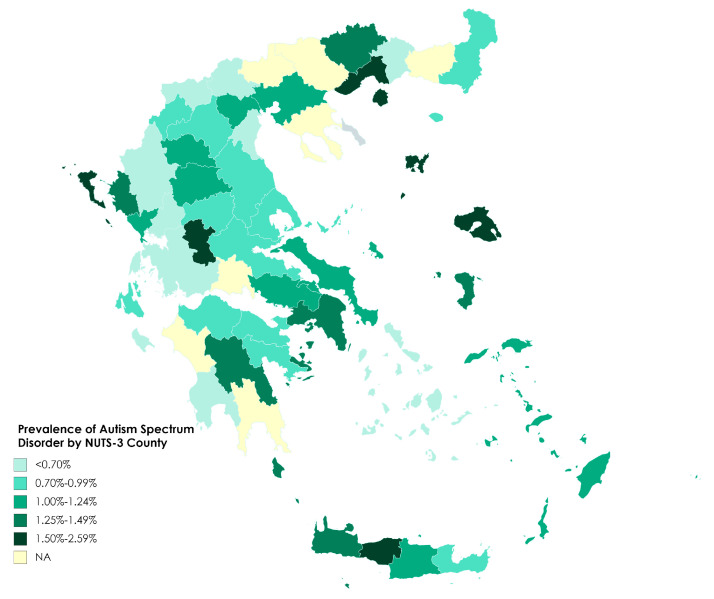
Prevalence of Autism Spectrum Disorder in areas covered by each Center for Educational Counseling and Support (corresponding to the European Union’s NUTS-3 level of territorial classification). NA = not available.

**Table 1 jcm-09-02163-t001:** Prevalence of Autism spectrum disorders (ASD) diagnosis in Greece in 2019 in children born in 2008 or 2009, by region.

Administrative Region	N *	ASD **	
		*n*	%
North Aegean	3070	46	1.50
Attica	57,186	837	1.46
Ionian Islands	4606	64	1.39
Crete	15,052	203	1.35
Eastern Macedonia and Thrace	9920	121	1.22
Central Macedonia	31,072	317	1.02
Central Greece	9636	97	1.01
Thessaly	14,372	133	0.93
Peloponnese	9160	83	0.91
South Aegean	6835	58	0.85
Epirus	6107	48	0.79
Western Macedonia	5149	38	0.74
Western Greece	10,714	63	0.59
Total Greece	18,2879	2108	1.15

* Population of children in the areas served by the participating centers whose year of birth was 2008 or 2009. ** Autism Spectrum Disorders.

**Table 2 jcm-09-02163-t002:** Prevalence of ASD diagnosis in 2019 in children born in 2008 or 2009, by gender and region.

Administrative Region	Males			Females			M/F Ratio
	N *	ASD **		N *	ASD		
		*n*	%		*n*	%	
North Aegean	1567	42	2.68	1503	4	0.27	10.1
Attica	29,349	672	2.29	27,837	165	0.59	3.9
Ionian Islands	2418	52	2.15	2188	12	0.55	3.9
Crete	7651	169	2.21	7401	34	0.46	4.8
Eastern Macedonia & Thrace	5074	91	1.79	4846	30	0.62	2.9
Central Macedonia	15,843	257	1.62	15,229	60	0.39	4.1
Central Greece	5024	78	1.55	4612	19	0.41	3.8
Thessaly	7371	115	1.56	7001	18	0.26	6.1
Peloponnese	4703	62	1.32	4457	21	0.47	2.8
South Aegean	3464	48	1.39	3371	10	0.30	4.6
Epirus	3210	41	1.28	2897	7	0.24	5.3
Western Macedonia	2663	32	1.20	2486	6	0.24	5.0
Western Greece	5560	56	1.01	5154	7	0.14	7.4
Total Greece	93,897	1715	1.83	88,982	393	0.44	4.14

* Population of children in the areas served by the participating centers whose year of birth was 2008 or 2009. ** Autism Spectrum Disorders.

**Table 3 jcm-09-02163-t003:** Autism cases: frequencies (%) by year of diagnosis up to the 10th year after the child’s birth and by gender.

Year (Mean Age) *	School Grade	Males %	Females %	Total%	Cumulative		
					Males %	Females %	Total %
3	-	3.7	4.1	3.8	3.7	4.1	3.8
4	Pre-school	13.2	14.6	13.5	16.9	18.7	17.3
5	Kindergarten	24.4	30.3	25.5	41.3	49.0	42.7
6	1st	21.1	20.3	20.9	62.3	69.2	63.6
7	2nd	16.6	15.9	16.5	78.9	85.1	80.1
8	3rd	9.3	6.2	8.7	88.2	91.3	88.8
9	4th	7.7	5.4	7.3	95.9	96.7	96.1
10	5th	4.1	3.3	3.9	100	100	100
	Total	100	100	100			

* Diagnoses were recorded by calendar year, not by the child’s actual age. See text for correspondence to range of child’s age.

**Table 4 jcm-09-02163-t004:** Age-specific prevalence of ASD diagnosis up to the 10th year after the child’s year of birth.

Year (Mean Age) *	School Grade	Males %	Females %	Total%	
					Male-to-Female Ratio
3	-	0.07	0.02	0.04	3.73
4	Pre-school	0.30	0.08	0.20	3.71
5	Kindergarten	0.74	0.21	0.49	3.46
6	1st	1.12	0.30	0.72	3.70
7	2nd	1.42	0.37	0.91	3.81
8	3rd	1.59	0.40	1.01	3.97
9	4th	1.73	0.42	1.09	4.07
10	5th	1.80	0.44	1.14	4.11

* Diagnoses were recorded by calendar year, not by the child’s actual age. See text for correspondence to range of child’s age.

**Table 5 jcm-09-02163-t005:** Mean age at ASD diagnosis and modal year of diagnosis after the child’s birth, by region.

Region	Number Diagnosed	Mean Age at Diagnosis	Modal Year
Epirus	48	5y2m	5
Western Macedonia	38	5y3m	4
North Aegean	46	5y4m	5
Attica	837	5y7m	5
Peloponnese	83	5y11m	5
Crete	203	6y2m	6
Thessaly	133	6y2m	5
E Macedonia & Thrace	121	6y3m	5
Central Greece	97	6y7m	7
South Aegean	58	6y9m	6
Ionian Islands	64	6y10m	7
Central Macedonia	317	7y0m	6
Western Greece	63	7y0m	7
Total Greece	2108	6y1m	5

**Table 6 jcm-09-02163-t006:** Year of ASD diagnosis up to the 10th year after the child’s birth, by region.

Region	Cumulative Frequency (%) by Year of ASD Diagnosis							
	3	4	5	6	7	8	9	10
Epirus	10.4	31.3	66.7	87.5	91.7	100	100	100
Western Macedonia	7.9	44.7	65.8	79.0	86.8	97.4	97.4	100
North Aegean	20.0	33.3	55.6	75.6	91.1	93.3	100	100
Attica	4.1	21.3	53.8	76.4	90.2	95.2	99.2	100
Peloponnese	6.0	16.9	53.0	71.1	83.1	86.8	94.0	100
Crete	3.1	20.9	38.3	60.2	79.6	86.2	95.4	100
Thessaly	2.3	14.1	41.4	60.2	81.3	91.4	93.8	100
E Macedonia & Thrace	2.5	17.8	41.5	58.5	74.6	83.1	94.1	100
Central Greece	3.1	8.3	19.8	43.8	76.0	90.6	97.9	100
South Aegean	0	3.5	25.9	53.5	67.2	84.5	94.8	100
Ionian Islands	3.2	6.5	24.0	42.9	59.0	74.5	91.2	100
Central Macedonia	1.0	6.6	24.0	42.9	59.0	74.5	91.2	100
Western Greece	5.5	9.1	20.0	30.9	61.8	83.6	90.9	100
Total Greece	3.8	17.3	42.7	63.6	80.1	88.8	96.1	100

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
