# Peer review of "Autism Spectrum Disorders in Greece: Nationwide Prevalence in 10–11 Year-Old Children and Regional Disparities"

_jcm, 2020, doi:10.3390/jcm9072163_

Round 1
Reviewer 1 Report
The manuscript “Autism Spectrum Disorders in Greece: Nationwide 2 prevalence in 10-11 year old children and regional 3 disparities” is interesting, comprehensive, well-written and well-structured.
I have a few minor comments and questions.
Introduction:
Page 1: The authors say that Autism Spectrum Disorders (ASD) are lifelong neurodevelopmental disorders of poorly defined etiology. I am a little hesitant to put it that way – poorly defined etiology? Autism, like other neurodevelopmental disorders, has numerous known pre- and perinatal etiologies but in a large number of cases the definite cause cannot be identified. There are several genetic syndromes associated with autism, with and without associated intellectual disability, several known prenatal exposures and perinatal risks, including very and extremely preterm birth. However, in only about 20-25% of cases a defined etiology can be ascertained.
Page 2: The statement that there is a marked gender difference with males more frequently affected than females could possibly be supplemented by mentioning that among children with autism and moderate to severe intellectual disability, the gender difference ratio is lower, in the range of 2:1 (boys:girls), see references by Fombonne 1999 and 2005.
Page 3: The authors give a valuable classification of methodological categories used in epidemiological studies.
Population and Methods:
Clear and well described.
It would have been very interesting if the authors could have given data on the children´s intellectual level, at least a rough estimate of the proportion with intellectual disability and with no intellectual disability. However, I understand that this was not possible due to available data and the authors have mentioned that as a limitation in the Discussion.
Results:
Well presented with also illustrations.
Discussion:
Also a comprehensive and informative discussion with a valuable overview of the literature and with comparisons with other countries in the world.
Page 15: The authors discuss that the proportion of diagnoses made up to and including the third year was particularly low in their studied population. They discuss several factors that may have contributed. On the following page, 16, screening for autism is discussed. Maybe, somewhere here the authors could mention the often very typical early sign of autism; delayed speech and language development.
Overall, the article provides the reader with an interesting and valuable overview of autism prevalence in Greece and in many parts of the world where epidemiological studies have been carried out.
Author Response
We very much appreciate the efforts of your referees and yourself in reviewing our submission. Our revised version includes several changes, which appear using Track Changes in the text.
In detail, we have responded to the referee’s comments as follows. Please note that only the comments that suggested action are listed. Naturally, we are very grateful for all the complimentary remarks.
Reviewer 1
Comment: Page 1: The authors say that Autism Spectrum Disorders (ASD) are lifelong neurodevelopmental disorders of poorly defined etiology. I am a little hesitant to put it that way – poorly defined etiology? Autism, like other neurodevelopmental disorders, has numerous known pre- and perinatal etiologies but in a large number of cases the definite cause cannot be identified. There are several genetic syndromes associated with autism, with and without associated intellectual disability, several known prenatal exposures and perinatal risks, including very and extremely preterm birth. However, in only about 20-25% of cases a defined etiology can be ascertained.
Reply: As suggested, we have deleted “of poorly defined etiology” from lines 31-32. In its place, we have added at line 45, following our brief mention of risk factors, “In most cases, the etiology remains unknown”.
Comment: Page 2: The statement that there is a marked gender difference with males more frequently affected than females could possibly be supplemented by mentioning that among children with autism and moderate to severe intellectual disability, the gender difference ratio is lower, in the range of 2:1 (boys:girls), see references by Fombonne 1999 and 2005.
Reply: We have made suitable additions in the Introduction and the Discussion to our existing comments on the gender difference. At lines 61-62, we added “The gender difference appears to be smaller among children with moderate to severe intellectual diability {3, 11]” – these references are to the two papers by Fombonne indicated by the Reviewer. In the Discussion, we added at line 387 the word “overall” to “Our overall male-to-female ratio of 4.14:1” for emphasis, and then added at lines 410-414 “In common with many studies, we were unable to examine whether the male-to-female ratio varied according to the level of intellectual functioning; there is evidence in the literature that it may be relatively low, around 2:1, among children with ASD and moderate to severe intellectual disability [3,11]”
Comment: Page 15: The authors discuss that the proportion of diagnoses made up to and including the third year was particularly low in their studied population. They discuss several factors that may have contributed. On the following page, 16, screening for autism is discussed. Maybe, somewhere here the authors could mention the often very typical early sign of autism; delayed speech and language development.
Reply: Our text already contains the following remark earlier on the same page “children with prominent symptoms such as language regression and more severe presentation, with autistic disorder and intellectual disability, are diagnosed earlier” (now at lines 421-423). So we respectfully suggest that it would be awkward and unnecessary to return to the subject, especially since we have not recorded the presenting symptoms.
Reviewer 2
Comment: Introduction: It is informative, clear and well written. If possible, I suggest to add the following reference in your paper: Epidemiology of Autism Spectrum Disorders: A Review of Worldwide Prevalence Estimates Since 2014. Chiarotti F, Venerosi A. Brain Sci. 2020 May 1;10(5):274.
Reply: The paper suggested by the Reviewer is particularly interesting because it investigates how prevalence estimates depend upon the methodology of the studies, which is a topic in our Discussion. Therefore, we propose referring to it in the Discussion instead of the Introduction, and correspondingly we have added at lines 384-385 “The association between prevalence estimates and source of information is investigated in a review of recent prevalence studies [67]”, where the reference is to the paper cited by the Reviewer.
Please note that following the insertion of this additional reference [67] in our Bibliography, the previous references 67- 93 necessarily become 68-94. We have made all these changes in the text and in the references but not marked them.
Please see the attached revised article.
We hope that you find these amendments satisfactory. Thank you again for your attention to our work.
Yours sincerely

Reviewer 2 Report
Dear Authors, I appreciated your contribute. It adds useful information in the field of epidemiology of ASD. Introduction: It is informative, clear and well written. If possible, I suggest to add the following reference in your paper: Epidemiology of Autism Spectrum Disorders: A Review of Worldwide Prevalence Estimates Since 2014. Brain Sci. 2020 May 1;10(5):274. Population and Methods: It describe in an operational way the steps of your research. Results: They are clearly represented. The tables and figures were well done. Discussion section is well supported by results.
Author Response

(The authors gave the same response as above.)
